# Endoglin: An ‘Accessory’ Receptor Regulating Blood Cell Development and Inflammation

**DOI:** 10.3390/ijms21239247

**Published:** 2020-12-03

**Authors:** Steffen K. Meurer, Ralf Weiskirchen

**Affiliations:** Institute of Molecular Pathobiochemistry, Experimental Gene Therapy and Clinical Chemistry (IFMPEGKC), RWTH University Hospital Aachen, D-52074 Aachen, Germany

**Keywords:** Endoglin, CD105, TGF-β-signaling, hematopoietic stem cells, hematopoiesis, innate immunity, adaptive immunity

## Abstract

Transforming growth factor-β1 (TGF-β1) is a pleiotropic factor sensed by most cells. It regulates a broad spectrum of cellular responses including hematopoiesis. In order to process TGF-β1-responses in time and space in an appropriate manner, there is a tight regulation of its signaling at diverse steps. The downstream signaling is mediated by type I and type II receptors and modulated by the ‘accessory’ receptor Endoglin also termed cluster of differentiation 105 (CD105). Endoglin was initially identified on pre-B leukemia cells but has received most attention due to its high expression on activated endothelial cells. In turn, Endoglin has been figured out as the causative factor for diseases associated with vascular dysfunction like hereditary hemorrhagic telangiectasia-1 (HHT-1), pre-eclampsia, and intrauterine growth restriction (IUPR). Because HHT patients often show signs of inflammation at vascular lesions, and loss of Endoglin in the myeloid lineage leads to spontaneous inflammation, it is speculated that Endoglin impacts inflammatory processes. In line, Endoglin is expressed on progenitor/precursor cells during hematopoiesis as well as on mature, differentiated cells of the innate and adaptive immune system. However, so far only pro-monocytes and macrophages have been in the focus of research, although Endoglin has been identified in many other immune system cell subsets. These findings imply a functional role of Endoglin in the maturation and function of immune cells. Aside the functional relevance of Endoglin in endothelial cells, CD105 is differentially expressed during hematopoiesis, arguing for a role of this receptor in the development of individual cell lineages. In addition, Endoglin expression is present on mature immune cells of the innate (i.e., macrophages and mast cells) and the adaptive (i.e., T-cells) immune system, further suggesting Endoglin as a factor that shapes immune responses. In this review, we summarize current knowledge on Endoglin expression and function in hematopoietic precursors and mature hematopoietic cells of different lineages.

## 1. Structural and Functional Aspects of Endoglin

Endoglin is a class I, single-membrane spanning receptor with an apparent molecular weight of 95 kDa containing a short cytoplasmic and a modular extracellular domain. This domain contains attachment sites for *N*- and *O*-dependent glycosylation and ligand binding residues. The homo-dimeric protein is stabilized by multiple intermolecular disulfide bridges [1,2]. In case of Endoglin, defective *N*-glycosylation interferes with membrane localization similar to the transforming growth factor-β type II receptor (TβRII), as well as it impacts exosomal targeting of Endoglin [1,3,4]. Direct binding of Endoglin to TGF-β is assumed to only occur in complex with the signaling receptors TGF-β type I and type II [2,5,6]. The Endoglin C-terminus is a substrate for TGF-β-receptors, leading to serine/threonine phosphorylation, which regulates the interaction of Endoglin with those receptors [7]. In addition, the C-terminal domain is tyrosine phosphorylated by Src kinase impacting Endoglin’s trafficking and endothelial responses [8]. In contrast to the signaling receptors, Endoglin possesses only a short intracellular domain with no intrinsic kinase activity. Nevertheless, this domain represents a hub for interaction with TGF-β/bone morphogenetic protein (BMP)-receptors, integrins and proteins regulating migration and trafficking such as zyxin, zyxin-related protein (ZRP)-1, β-arrestin2, and the GAIP C-terminus-interacting protein (GIPC) [9,10,11,12,13,14]. The interaction with GIPC affects in addition to Smads (see below) also PI3K/Akt and ERK1/2 activation (Figure 1) [12,14].

Beside the “regular” Endoglin variant (L-Endoglin), a shorter splice variant of Endoglin (S-Endoglin) has been identified [15,16,17]. This splice variant has a shortened C-terminus, which is missing phosphorylation and protein interaction sites, resulting in functional differences compared to L-Endoglin (see below). Upon ligand binding, TβRII and TβRI activation results in endocytosis of the receptor complex followed by phosphorylation of substrate proteins by TβRI. In general, there are two different ways of internalization involving clathrin-coated pits (ccp, early endosomal antigen (EEA)-1^+^) or alternatively caveolin-coated pits (lipid rafts, caveolin-1^+^). It is assumed that localization to ccp leads to Smad activation, whereas routing to lipid rafts leads to degradation of receptors or activation of non-Smad signaling [18,19]. In endothelial cells (EC), Endoglin is localized to the membrane of caveolae, where it regulates the stability and interaction of eNOS and its allosteric regulator Hsp90 [20]. In addition, Endoglin was found in detergent-insoluble proteins (representing most likely lipid rafts) in placentas of pre-eclamptic women [21]. Activated Smads translocate to the nucleus in complex with the common Smad4, and in concert with positive and negative transcription factors regulate a wide spectrum of target genes [22]. In ECs, Endoglin modulates diverse functions, including migration and proliferation, by differentially regulating Smad2/3 signaling versus Smad1/5/8 signaling, which involves the interaction of Endoglin with the structural protein GIPC [13,23]. Moreover, the phosphorylation of the C-terminal part of Endoglin generates docking sites for intracellular interacting proteins, which also applies to subsets of T-cells (see below) [24,25]. However, the intracellular signal transfer modalities to regulate transcription or influence cellular functions by Endoglin (at least in EC) are not restricted to Smad-dependent mechanisms [26,27].

Endoglin expression is upregulated in activated ECs and this is facilitated by different transcription factors, including TGF-β1 regulated Smad3/Sp1/KLF6 and hypoxia regulated Hif1α [28,29]. Due to the high expression of Endoglin in activated ECs, the function and biology of Endoglin has been analyzed in detail in ECs and processes based on angiogenesis including carcinogenesis [30]. Especially, Endoglin has been exploited to target anti-cancer drugs to the tumor vasculature to treat cancer [31,32]. In contrast to EC, in tumor cells of different origins, in which Endoglin acts as a tumor suppressor, its expression is epigenetically silenced/switched off [33,34,35,36].

The area of EC research was enforced by the finding that the human disease hereditary hemorrhagic telangiectasia 1 (HHT-1) is based on genetic defects in the *endoglin* gene (*Eng*), leading to defect/non-functional or mis-targeted proteins causing haploinsufficiency in Endoglin [1,37,38,39,40]. Several Endoglin mutants have been shown to get stuck in the endoplasmic reticulum due to folding/glycosylation defects [41]. Those proteins are most likely ubiquitinated, a fact substantiated by the interaction with the E3 ubiquitin ligase tripartite motif-containing protein 21 (TRIM21) [42]. Likewise, to the multiple and sometimes life-threatening complications in HHT-1 patients, the loss of Endoglin causes embryonic lethality in homozygous Endoglin knockout mice, underpinning the important function of Endoglin during development [43]. More importantly, Endoglin knockout mouse models revealed that not only EC but also vessel-associated cells express Endoglin, and the communication of these cells with EC is the basis for defective angiogenesis [44]. In addition, the function of Endoglin is not limited to Endoglin expressing cells, since this receptor itself can directly be a part of the paracrine communication by the proteolytical release of the extracellular domain as soluble Endoglin (sEng) from the cell surface [45]. sEng is increased in pre-eclamptic woman and has been functionally linked to this disease [46]. Moreover, sEng and the membrane bound full-length Endoglin can be released by cells as a cargo of exosomes [4].

In summary, these studies demonstrate that under physiological conditions (i) Endoglin is also functionally expressed outside ECs, (ii) Endoglin is a co-receptor primarily (but not exclusively) for ligands of the TGF-β-superfamily, and finally (iii) Endoglin not only impacts the function of cells expressing the receptor, but also acts in a paracrine manner.

## 2. Endoglin in Pathological Conditions

As discussed, Endoglin is highly expressed in active/angiogenic ECs. Therefore, dysfunction of this receptor affects several organ systems relying on angiogenesis.

### 2.1. The Female Reproductive System

In the mouse, endoglin mRNA and protein could be localized to blood vessels and capillaries (EC) as well as interstitial fibroblasts in several tissues, [47]. Endoglin is highly expressed in the ovary, uterus and the cDNA has been cloned from a placenta library [47]. During placental development, the establishment of the fetal-maternal interaction is critical for successful pregnancy [48]. Abnormalities of placenta formation due to shallow trophoblast invasion have been linked to pre-eclampsia and fetal/intrauterine growth restriction (IUPR) [49]. Pre-eclampsia is characterized by the onset of hypertension and proteinuria in the third trimester of pregnancy and the severe forms can lead to the HELLP (Haemolysis, Elevated Liver enzyme levels, Low Platelet count) syndrome and IUPR [50,51]. The disease is based on excess placental-derived soluble VEGF receptor (sVEGFR1, sFlt1) as well as an excess of sEng released from endothelial membrane bound L-Endoglin, of which both, i.e., sFlt1 and sEng, are found elevated in the serum of pre-eclamptic women [46,52]. The liberation of sFlt1 and sEng prevents binding of their cognate ligands to EC, thereby affecting vascular tone and tissue oxygenation. In line, IUPR is often caused by hypoxia and is likely resulting from shallow trophoblast invasion in the decidua and myometrium and failure to invade the spiral arteries, a key requirement to establish an efficient utero-placental circulation [53].

In addition to EC, during pregnancy, Endoglin is permanently highly expressed on syncytiotrophoblasts and is also upregulated on extravillous trophoblasts differentiating along the invasive pathway [47,54,55]. The analysis of human villous explants revealed that artificial downregulation of Endoglin by an antibody or antisense oligonucleotides stimulates outgrows, migration and a higher Fibronectin release [56]. As mentioned above, IUPR is accompanied by placental hypoxia and increased TGF-β3 expression. Low oxygenation induces Endoglin expression (full-length and soluble form) mediated by TGF-β3 [57]. Aside ECs and trophoblasts, the expression of Endoglin was upregulated in *decidua basalis* of mid pregnancy in macrophages [58].

### 2.2. Vascular Homeostasis

The fact that HHT-1 is based on Endoglin mutations and HHT-2 is based on ALK1 mutations, both components of the TGF-β-pathway, account for ~85% of clinically diagnosed HHT patients implies that TGF-β is a critical component in this disease [59,60,61,62]. Comparative gene array analysis of human umbilical vein endothelial cells (HUVECs) from HHT-1 patients and control newborns showed several differentially expressed genes, comprising functions in extracellular matrix formation, angiogenesis, cellular adhesion and affecting genes involved in TGF-β signaling (i.e., Smad1, Smad7, and TGF-β2) [63]. Nevertheless, gene expression profiling of human nasal telangiectasial tissue and non-telangiectasial tissue revealed that not only TGF-β-related candidates are differentially expressed, but Wnt signaling also seems to be affected [62].

TGF-β 1 is a critical regulator in the endothelial system, but this applies also to hematopoiesis and mature hematopoietic cells. Therefore, the expression of Endoglin in the corresponding cells is of fundamental interest [64]. A first functional link of Endoglin to the immune system, i.e., a role of Endoglin in macrophage biology, was realized quite early. In particular, it has been noted that mononuclear cell infiltrates are observed around telangiectases consisting of lymphocytes and monocytes/macrophages [65,66]. However, immunophenotypic analysis of T-, B- and NK-lymphocytes indicated that there were no quantitative or qualitative abnormalities in HHT patients. There was no activation of T-cells and normal levels of IgGs, but the activity of components of the innate immune system was affected [67]. The phagocytotic activity and/or the oxidative burst of polymorphonuclear neutrophils (PMN) and/or monocytes were shown to be reduced in HHT patients [67]. In contrast, Guilhem and colleagues found no difference in innate immunity, but lymphopenia of T-cells (CD4, CD8) and NK cells increased, and the level of IgG and IgA was found elevated [68]. In another study, the adaptive immune system showed no quantitative change in B- and T-cells, but there was a deficit in T-cells expressing Th1-relevant cytokines (IFN-γ, IL2, TNF-α) and monocytes positive for TNF-α rendering HHT patients more susceptible to infections [69].

It is known that TGF-β1 induces macrophage differentiation [70]. Moreover, neutrophil survival, chemotaxis and activation (examined by oxidative burst and phagocytosis) are increased by TGF-β1 [71]. Therefore, it is obvious that immune cell dysfunction in the absence of Endoglin might be linked to TGF-β1, a fact substantiated by the lower TGF-β1 plasma level found in heterozygous Endoglin deficient mice compared to wild-type mice [72]. A more indirect role of Endoglin in ECs with respect to inflammation is its involvement in cell-cell contacts governing vessel architecture and the recruitment of immune cells (all lineages) from the circulation (extravasation) [44,73,74]. In turn, increased expression of Endoglin, which occurs for example in the autoimmune thyroiditis Graves’ disease and psoriatic lesions, might lead to an enhanced inflammatory cell recruitment [75,76].

## 3. Endoglin in the Hematopoietic System

### 3.1. Endoglin Expression in the Stem Cell/Niche

In the follow-up of radio-/chemotherapy, there is routinely performed a hematopoietic stem cell transplantation to reconstitute the hematopoietic system of the treated cancer patients. To circumvent problems of neutropenia and thrombocytopenia as a short-term consequence of transplantation, mesenchymal (MSC)/hematopoietic stem cell (HSC) co-transplantation is often used [77]. MSCs represent a population of multipotent somatic stem cells present in various tissues including bone marrow [78]. Their phenotype is characterized by expression of Endoglin amongst other markers. These cells were used to generate the monoclonal antibody SH-2 directed against Endoglin [79,80]. MSC are a promising source of starting material for tissue engineering, cellular and somatic gene therapy and do possess immunomodulatory features [81]. In order to achieve amplification of MSC (Stro^+^/CD34^−^) and HSC (Stro-/CD34^+^) in sufficient amounts, a bioreactor procedure was established [82]. Starting from mononuclear cells isolated from patients’ bone marrow, both cells were amplified alongside [82]. The expanded MSCs are positive for mesenchymal progenitor cell markers, including vimentin and Endoglin, but prove to be negative for lineage specific markers, including Collagen type II (chondrogenesis), osteocalcin (osteogenesis) and C/EBPα (adipogenesis). Treatment of the derived MSC using appropriate differentiation conditions, chondrocytes, osteocytes and adipocytes could be generated [82]. Recently, there has been identified an Endoglin negative subpopulation of murine adipose tissue-derived MSC (mASC), which are more efficient in inhibiting T-cell proliferation when compared to the Endoglin expressing counterparts [83].

The postnatal mammalian bone marrow compartment is the side where hematopoiesis and osteogenesis take place. In addition to the cartilaginous, osteoid and hematopoietic lineages, the hematopoietic niche cells are present in this compartment. This diverse set of cells supports HSC to generate all lineages of blood and immune cells [84]. Chan and colleagues isolated a lineage–restricted, self-renewing common skeletal progenitor (termed bone cartilage, stromal progenitor; BCSP) from limb bone marrow of fetal, neonatal and adult mice [85]. The BCSP clonally produces chondrocytes, osteogenic cells and three subsets of stromal cells, which exhibit differential expression of markers including CD105. Fluorescence-activated cell sorting (FACS) analysis of limb progenitors at different developmental stages indicated that CD105 is an early marker of skeletal lineage commitment from embryonic day (E)13 on. [85]. Moreover, isolated CD105^+^ skeletal progenitor cells give rise to bone, chondrocytes and multiple types of osteogenic stroma cells (multipotent BCSP). A subset of CD105^+^ cells has the potency to promote the survival and maintenance of multi-lineage reconstitution by HSC, whereas another CD105^-^ BCSP-derived lineage supports or directs B-lymphopoiesis exclusively [85].

### 3.2. Endoglin Expression During Hematopoiesis

#### 3.2.1. Fetal Hematopoiesis

In detail (in vivo) it was shown that Endoglin is co-expressed with GATA1 in primitive erythroid progenitors (EryPs), which are significantly reduced in *Eng*^−/−^ embryos. At E9.5 a reduction of erythroblasts (CD71^+^/Ter119^+^) in the yolk sacs of *Eng*^−/−^ embryos accompanied by a lower β-globin expression confirmed impaired primitive erythropoiesis in the absence of *Eng* [86]. Co-expression of Endoglin with fetal liver kinase-1 (Flk-1) is detected in the primitive streak (PS) in mesodermal cells of blood islands at E7.5 [87]. FACS analysis revealed that only Endoglin expressing cells are endowed with hematopoietic potential at E7.5. Furthermore, only Eng^+^ and Flk-1^+^ cells show hematopoietic and endothelial cell potential, i.e., hemangioblast or Blast-colony forming cells (BL-CFC) [86]. In line, the quantification of the hematogenic progenitor activity at E9.5 revealed significantly reduced numbers of erythroid burst-forming unit (BFU-E) in *Eng*^−/−^ yolk sacs, but no effect on the granulocyte-macrophage colony-forming unit (CFU-GM) lineage, confirming that Eng affects definitive hematopoiesis and those effects might be regulated by BMPs or TGF-β1 and corresponding downstream target genes [86] (Figure 2A). Using the ES cell system and in vitro differentiation of hematopoietic lineages, it could be shown that CD105 is already co-expressed with Flk1^+^ in precursors with hematopoietic potential and still detectable in the earliest CD45^+^ cells [88]. Deficiency of CD105 (*Eng*^−/−^) in ES cells, did not affect the progression to the Flk-1^+^ stage of mesenchymal precursors (hemangioblasts), representative for definitive hematopoiesis [88]. However, the CD45^+^ subset of hematopoietic cells is reduced at d9. In addition, the differentiation of ES cells into erythroid (CD4^−-/^ER119^+^) and myeloid (CD45^+^/CD11b^+^) was severely diminished along with a strongly reduced adult β-globin gene expression indicating impaired definitive erythropoiesis. These data indicate that myelopoiesis and definitive erythropoiesis were severely impaired, leaving the lymphoid lineage unaffected [88].

In order to shed more light on the function of Endoglin during the steps of primitive hematopoiesis (yolk sac hematopoiesis), Perlingeiro and colleagues employed the ES cell system—using ES *En*g^−/−^ cells - to analyze their hemangioblast activity and their hematopoietic and endothelial potential [89]. They found that Endoglin is already expressed in ES cells and represents together with Flk-1 a marker for BL-CFC (hemangioblast). Endoglin deficiency reduces the number and size of BL-CFC as well as their hematopoietic potential (erythropoiesis), which is compatible with the anemia seen in *Eng*^−/−^embryos [87,89]. In contrast, the endothelial developmental potential is even promoted in the absence of Endoglin in BL-CFC, whereas the branching ability in a sprouting assay is reduced similar to that seen in *Eng*^−/−^ embryos [89,90]. These results were corroborated in ES cells with doxycycline-mediated overexpression of Endoglin causing a higher number of BL-CFC. In addition, these BL-CFCs had a reduced endothelial potential, but a higher erythroid potential evident as elevated erythroid colony-forming progenitor colonies and BFU-E. Myeloid progenitors were unaffected underpinning the role of Endoglin in erythroid development [91]. Downstream of Endoglin, Smad1 and the master hematopoietic regulator *Scl* (stem cell leukemia) were identified to mediate the observed effects of CD105 on BL-CFC and their erythroid/endothelial potential [91]. In more detail it was shown that the differential effect of Endoglin on hematopoiesis and endothelial cell potential/cardiac differentiation is mediated by a crosstalk of BMP and Wnt signaling [92] (Figure 2B).

In terms of the fetal hematopoiesis it can be summarized that Endoglin is expressed in early erythroid precursors (EryP) and mesodermal precursors with hematogenic and endothelial potential. Aberrant expression of Endoglin affects mainly the erythroid lineage but also to a minor extent the myeloid and lymphoid lineages.

#### 3.2.2. Adult Hematopoiesis

In the adult bone marrow, Endoglin (CD105) and the signaling lymphocyte activation molecule 1 (SLAMF1, CD150) are two cell surface receptors highly expressed in the primitive HSC KLS^+^ (Kit^+^/Lin^−^/Sca-1^+^) compartment [93,94]. In this compartment, Endoglin is a functional marker that defines long term repopulating HSC (LT-HSC, CD34^−^/Flk2^−^) [93,95]. The highest Endoglin expression is found in long-term HSC (CD34^−^), whereas CD105 expression in short-term HSC (ST-HSC, CD34^+^/Flk2^−^) and multipotent progenitors (MPP, CD34^+^/Flk2^+^) is much lower [93,95] (Figure 3A). In the KLS^−^ (Kit^+^/Lin^−^/Sca-1^−^) primitive progenitor compartment, the common lymphoid progenitor and the common myeloid progenitor show only very low CD105 expression [94]. In contrast in the committed precursor cells, i.e., the megakaryocyte-erythroid progenitor (MEP), Endoglin expression is high compared to the low expression in the granulocyte-monocyte progenitor (GMP) [94]. Using a different FACS marker panel led to a higher resolution of the myeloid lineage, indicating that, in the myelo-erythroid progenitor compartment Endoglin expression is high, primarily in the erythroid committed pre-CFU-E and CFU-E progenitors [94,96,97] (Figure 3B).

With respect to the mature cells circulating in the peripheral blood (Figure 3C) as well as cells which mature in the destination tissue, i.e., mast cells and macrophages, (Figure 3D), only erythrocytes, thrombocytes and granulocytes have not been shown to express Endoglin so far. Initial studies on circulating human CD34^+^ primitive progenitors, which lack any differentiating antigen, showed that a subfraction of these cells expresses higher CD105 (20–38%) and those cells were CD25^+^. This fraction has a delayed proliferative response, contains the majority of long-term culture-initiating cells (LTC-IC), and multipotential colony-forming cells [98,99]. Further analysis revealed that CD105^+^ cells were virtually devoid of GM-CFU and BFU-E, whereas megakaryocytic aggregates with cloning potential were retained exclusively in this progenitor population. In addition, these cells had a higher secondary colony-forming potential, including GM-CFU-E, BFU-E and megakaryocyte aggregates [99]. Upon treatment of CD105^+^ cells with an anti-TGF-β1 antibody, respective cells increased colony-forming activity (GM-CFU-E, BFU-E), implying that CD105^+^ cells contain committed, directly clonable progenitors which are kept quiescent by autocrine activity of TGF-β1 [99] (Figure 3E).

In order to evaluate the functional impact of Endoglin on bone marrow progenitor cells in the adult, bone marrow cells were treated with 5-Fluorouracil (5-FU), which blocks cell cycle in progenitor cells, leads to enrichment of rarely cycling cells expressing high quantities of Endoglin [100]. Those cells were either transduced with a lentiviral shRNA construct leading to a reduction of CD105 (both splice variants) or alternatively with a vector encoding L-Endoglin causing overexpression of this Endoglin isoform [100]. These experiments implied that Endoglin itself does not affect the clonogenic/proliferative potential of hematopoietic progenitors but reduced the anti-proliferative effect of TGF-β1 on myeloid progenitors [100]. Furthermore, it was shown that Endoglin expression in HSC has no impact on the engraftment and reconstitution potential of HSC in the transplantation setting, resulting in similar amounts of T-cell (CD3^+^), B-cell (B220^+^) and macrophage (Mac-1^+^) lineages. However, when recipient bone marrow was analyzed post-transplantation by FACS isolation of GFP^+^ (Lin^−^/Kit^+^/Sca-1^+^) cells, it turned out that the frequency of BFU-E is increased in the absence of Endoglin. In addition, overexpression of Endoglin decreased basophilic erythroblasts (CD71^+^/Ter119^+^). These data imply that Endoglin negatively regulates proliferation of BFU-E progenitors and impairs the erythroid differentiation at the basophilic erythroblast stage, without changing the red blood cell count (RBC) of recipients [100] (Figure 3F).

However, phenotypically characterization of different subsets of blood cells showed that in general the normalized mRNA expression of *endoglin* in different peripheral blood cells is not necessarily congruent. Very low expression is seen in granulocytes, whereas highest expression is observed in the different monocyte subtypes. In T- and B-cells *endoglin* mRNA is comparable low expressed (Figure 4).

Consequently, in adult hematopoiesis Endoglin expression is high in long term HSC in the KLS^+^ compartment and decreased in short-term HSC and MPP. In the KLS^−^ compartment, Endoglin is transiently high expressed in MEP, pre-CFU-E and CFU-E., while dysregulation of Endoglin expression affects mainly the erythroid lineage.

#### 3.2.3. Expression of Endoglin in Individual Hematopoietic Lineages

##### Erythrocytes

Initial studies of sorted bone marrow mononuclear cells revealed that Endoglin is expressed on immature pro-erythroblasts using the Endoglin-specific 1G2 antibody [102]. In the normal adult bone marrow, Endoglin is expressed in 3–5% of mononuclear cells, which are glycophorin A^+^ and display a pro-erythroblast phenotype [102]. Della Porta and colleagues isolated bone marrow erythroblasts using the markers LDS751^+^/CD45^−^/GlyA^+^ and these cells co-expressed CD105 [103]. In this study, it could be shown that CD105 expression is increased in bone marrow erythroid cells of patients with the myelodysplastic syndrome [103,104]. Since a distinctive sign of ineffective erythropoiesis is a high proportion of immature erythroblasts, CD105 could be included in a marker panel to identify erythroid dysplasia in those patients suffering from this myelodysplastic syndrome [103,104].

Endoglin, VE-cadherin, CD31, Tie2, and CD34 are established markers for embryonic and adult ECs [105,106,107,108]. In a study by Ema and colleagues, it was shown that cells positive for these EC markers arise from a subset of Flk1^+^ (VEGF-R2) mesodermal cells [109]. A subset of these Flk1^+^ cells express GATA1, a marker for primitive erythropoietic progenitor cells. Accordingly, they concluded that in addition to definitive also primitive hematopoietic cells arise from EC marker positive cells (including Endoglin, see above) [109,110]. A role for Endoglin in the BFU-E and erythroid differentiation was shown in transplantation experiments using bone marrow cells with modified Endoglin expression [100]. Analysis of a subset of recipient bone marrow cells by enrichment of BFU-E progenitors (KLS^+^/CD150^+^) revealed that Endoglin has a negative impact on early erythroid proliferation at the BFU-E stage and erythroid differentiation at the erythroblast stage (see above) [100].

Diamond–Blackfan anaemia (DBA) is an inherited bone marrow failure syndrome with the hematologic hallmarks of normochromic, macrocytic anemia and reticulocytopenia, which stems from a paucity of erythroid precursors in the bone marrow [111]. In animal models it was shown that the red cell aplasia in DBA arises from a defect in erythroid progenitors with a reduced ability of DBA bone marrow to generate BFU-E and erythroid colony-forming unit (CFU-E). The authors of this study defined two individual erythroid progenitors, i.e., early erythroid progenitors (Lin^−^/CD34^+^/CD38^+^/CD45RA^−^/CD123^−^/CD105^−^/CD36^+^) giving rise to BFU-E, and late erythroid progenitors (Lin^−^/CD34^+/−^/CD38^+^/CD45RA^−^/CD123^−^/CD105^+^/CD36^+^) giving rise to CFU-E. Only CFU-E expressed transiently Endoglin (until day 11) [111]. The study revealed qualitative and quantitative defects in these two progenitor fractions leading to a paucity of erythroblasts. In another study by Mori et al. 2015, it was shown that the megakaryocyte erythrocyte progenitor fraction (hMEP) in humans can be subdivided by the presence or absence of CD71 and Endoglin in three subfractions [112]. Those cells expressing CD71 intermediate (CD71^int^) and Endoglin were completely restricted to the erythroid lineage. Application of TGF-β1 to all three subfractions decreased proliferation and promoted erythroid terminal differentiation, an effect that was more pronounced in the presence of Endoglin [111]. In erythrocytes and respective precursor cells, Endoglin expression so far is only used to be included in FACS panels to determine the developmental stage of cells of the erythroid lineage affected by a certain disease, e.g., myeloplastic dysplasia or DBA. Conclusion: Due to its transient expression in precursors, Endoglin might be involved in the differentiation of erythroid cells, but there are no functional roles assigned to Endoglin in mature cells of this lineage.

##### Megakaryocytes/Thrombocytes

In transplantation experiments, using Endoglin expression modified bone marrow cells, there was no effect seen on platelet count in isolated plasma of recipient mice [98]. However, one of the characteristic signs of HHT is recurrent nosebleeds, which have been assigned to defects in the vascular system [113]. However, the coagulation time and prothrombin time in HHT patients is normal, despite showing a higher factor VIII level [114]. A recent study analyzed the interaction of thrombocytes and ECs in inflammatory conditions, a process which would affect the primary homeostasis. In this study endothelial Endoglin was shown to mediate thrombocyte adhesion via the thrombocyte integrin complex αIIbβ3 [115].

##### Granulocytes

With respect to granulocytes, there are no data showing expression or functional impact of Endoglin on this cell lineage. In line, there was no immunoreactivity with the antibody 44G4 of isolated human peripheral blood granulocytes [116]. Analysis of bone marrow with the antibodies 44G4 and 1G2 showed only reactivity with pro-erythroblasts but not with T-cells, B-cells, NK-cells or myeloid cells [102]. FACS analysis of bone marrow cell populations showed that in the committed cells, including erythroid (Ter119^+^) and B-lineage (B220^+^), endoglin was lowest expressed in myeloid cells (Mac-1^+^/Gr-1^+^) [100]. In inflammatory conditions using the zymosan-induced peritonitis model, there was no Endoglin expression detected in granulocytes by FACS analysis of myeloid cells [117].

##### Monocytes and Macrophages

In general, the increase in endoglin expression in different human inflammatory settings was found to be not restricted to ECs but associated with the macrophage/T-cell infiltrate [118]. Early studies led to the identification of Endoglin in the pro-monocytic cell line U937 in addition to the pre-B-leukemia cell line HOON and HUVECs [54,116,119,120]. Similar to the granulocytes, monocytes isolated from peripheral blood were found to be negative for Endoglin (44G4/8E11). Nevertheless, Endoglin is expressed by in vitro differentiated monocytes [121] (Figure 5A). Endoglin was also found in interstitial macrophages present in the red pulp of the spleen, while liver resident macrophages (Kupffer cells), tonsil, lymph node, and lung were negative [121]. Expression of Endoglin (8E11) is strongly increased by phorbol-12-myristate-13-acetate treatment (PMA) triggering differentiation in U937 (pro-monocytic) and HL-60 (myelomonocytic) cells [121] (Figure 5B). In line, O′Connell and colleagues generated an antibody RMAC8 by immunization mice with HUVECs [122]. Using RMAC8, low Endoglin expression was detected on primary monocytes, and human leukemic cell lines HL-60 and U937. Marginal expression is also evident in different pre-B-leukemia cell lines. Higher expression was detected in macrophages, but granulocytes, T-cells, B-cells and corresponding cell lines lack Endoglin. During PMA-induced differentiation of monocytes to macrophages Endoglin is upregulated [122]. Differentiation/polarization of monocytes increases Endoglin expression, which is detected in both M1 and M2 macrophages, respectively [117] (Figure 5A).

A reduced inflammatory response including less abundance of macrophages was seen in a murine renal ischemia-reperfusion injury and irradiation model in Eng^+/−^ compared to wild-type mice, implying Endoglin in macrophage recruitment/maturation [123,124]. Endoglin expression was also elevated in macrophages in the process of rheumatoid arthritis in synovial tissue compared to osteoarthritis or normal synovial tissue as well as in autoimmune thyroid disorders [75,125]. Similar to the data of O′Connell [122], Wang and colleagues found no expression in the immortalized erythroblastoid cell line K-562, the lymphoblastoid cell line JY, and the T-cell line Jurkat. Nevertheless, upon transient overexpression of progressive ankylosis (*Ank*) in K-562 cells, which shifts the cells to the early stages of erythroid differentiation, Endoglin is expressed [126].

In a more recent study, it was shown that the secretome of GM-macrophages (M1 polarized, pro-inflammatory) acts anti-angiogenic by inhibiting tubulogenesis and EC migration, an effect not seen by using the secretome of M-macrophages (M2 polarized, anti-inflammatory) [127]. In their study, the authors could show that elastase (MMP-12) is highly (functionally) expressed in GM-macrophages but not in M-macrophages and is the mediator of the anti-angiogenic activity in the secretome of GM-macrophages. Along with the MMP-12 expression, the presence of soluble Endoglin (sEng, anti-angiogenic) is detected in the supernatants of GM-macrophages in a much higher amount compared to M-macrophages [127]. sEng is derived from the membrane bound L-Eng through a shedding mechanism mediated by MMP-14 [45]. In ECs, sEng has a pro-inflammatory and anti-angiogenic function [46,128]. Moreover, sEng modulates monocyte adhesion and transmigration [73]. In afore mentioned setting, it could be demonstrated that GM-macrophage derived MMP-12 sheds Endoglin from macrophages and ECs *in vitro* and *in vivo* [127] (Figure 5C).

Aside the initially identified long form of human Endoglin (L-Eng), a short splice variant S-Eng was identified in ECs, placenta and myelomonocytic cell lines (HL-60, U937), both of which bind TGF-β1 [129]. S-Eng is also expressed in mouse and rat and due to its different C-terminus compared to L-Eng acts in a different way in processes of differentiation, cancer, migration and senescence [16,17,130,131,132,133]. To shed more light on the function of Endoglin in monocytes and the differential roles of L- and S-Eng, RNA of corresponding U937 were analyzed by using a genomic expression microarray [130,133]. These studies revealed that key processes are affected including cell adhesion and transmigration, which are crucial for leukocyte recruitment in an inflammatory scenario [133]. Both variants of Endoglin downregulated the chemokine receptor CCR2, and different integrin subunits, an effect which could be verified in peripheral blood macrophages from HHT-1 patients and Eng targeted, siRNA-treated HUVECs [133]. Impaired homing of peripheral blood mononuclear cells has been described for HHT-1 patients before [134]. On the other hand, the ligands Activin A (macrophage differentiation/polarization) and IL-1β were differentially regulated by the two isoforms of Endoglin. In addition, both Endoglin variants reduce the macrophage maturation marker CD11b [133]. Analysis of primary macrophages isolated from the peritoneal cavity of mice revealed that S-Eng is upregulated in senescent macrophages and in murine and human macrophages exposed to oxidative stress [135]. In order to differentiate the functional effects of L-and S-Eng, U937 transfectants with both isoforms were subjected to proteome analysis [135]. In combination with the performed functional analysis it can be concluded that S-Eng lowers proliferation, increases reactive oxygen stress, and impairs the monocytic differentiation into the pro-inflammatory M1 macrophage phenotype [135] (Figure 5D).

In a very recent study, the mechanism of intron retention (IR) on the development and activation of macrophages was analyzed [136]. IR is one of the modes of alternative splicing and describes the retention of an intron in the mature mRNA [137,138]. IR is now recognized as a key mechanism regulating gene expression also in the hematopoietic system [139,140]. The short splice variant, S-Eng, is generated by intron retention-mediated by the ASF/SF2 splicing factors during endothelial senescence [141]. In rat hepatic stellate cells there is also a shorter splice variant generated by intron retention with an altered C-terminus [17]. This splice variant differs in amino acid sequence from the mouse and human orthologues. If IR affects the function of rat endoglin is currently not known. With respect to the role of IR in macrophage development, it turned out that IR decreases, paralleled by an upregulation of genes (spliced mRNA) important for the monocyte-to-macrophage development [136]. Amongst these genes were Id2 and Endoglin. Less retention of intron 7 in Endoglin correlates with an increase in Endoglin mRNA expression. In addition, it was shown that retention of the mRNA including intron 14 in Endoglin is not subject to the cytoplasmic nonsense-mediated decay pathway but detained in the nucleus [142,143]. These accumulated nuclear transcripts can be processed by constitutive splicing, resulting in a rapid burst of protein synthesis [144]. Finally, a conditional knockout mouse approach of Endoglin in the myeloid lineage (Eng^fl/fl^/LysMCre) revealed a reduced immune response in respective mice with spontaneous infections mostly *Staphylococcus aureus* [117]. Moreover, these mice show a delayed lipopolysaccharide-induced mortality rate accompanied by reduced expression of the pro-inflammatory cytokines IL-6 and IL-1β. In addition, phagocytosis and TGF-β1 target gene expression, i.e., ALK-1, inducible NO-synthase (iNOS), MMP-12, was reduced in corresponding macrophages. In addition, less macrophage transmigration was observed *in vivo*. In line with the overexpression studies in U937, *Inhba* (*activin A)* expression was reduced (Figure 5E) [117,131].

Overall, these data imply that Endoglin is expressed and upregulated during macrophage differentiation and polarization. Endoglin most likely favors the pro-inflammatory M1 macrophage decision and Endoglin deficiency in macrophages renders an individual prone to infection.

##### Mast Cells

Mast Cells and TGF-β1

Similar to other immune cells, TGF-β1 controls a broad spectrum of MC functions, including proliferation, cell cycle control, and apoptosis [145]. Mast cells (MCs) express and secrete TGF-β1, which in turn regulates the expression of proteases and their release by down-regulation of MC-specific receptors controlling cellular degranulation, including KIT (ligand SCF) and FcεRI (ligand IgE), *in vitro* and *in vivo* by TGF-β1 in a SMAD-dependent fashion [146,147]. In addition, TGF-β1 suppresses IL-33-induced cytokine production and MC activation by interfering with MAP kinase phosphorylation [148]. Interestingly, the impact of TGF-β1 on chemotaxis is not mediated by the classical SMAD pathway, but by MEK1/2 signaling [149]. Moreover, the SRC family kinase FYN plays a critical role in TGF-β1-mediated MC migration *in vitro* and *in vivo* [150]. MCs express type I, type II and type III TGF-β receptors enabling them to bind this ligand [151]. Nevertheless, the effects are somewhat mixed, leading on one hand to chemotaxis and increased effector synthesis [151,152], and on the other hand apoptosis, blocking late stage maturation and sensor suppression [152,153,154].

Mast Cells and Endoglin

In most of the studies dealing with MCs and Endoglin, the effect of MCs on microvessel density (angiogenesis) was analyzed. Therefore, tryptase (MCs) and endoglin (ECs) were correlated [155,156,157,158,159]. Because MCs are located mostly close to vessels and ECs it is assumed that the increase in MCs observed under different pathological conditions stimulate *via* release of pro-angiogenic mediators EC activation. In a model of focal cerebral ischemia, it was shown that the breakdown of the blood brain barrier is reduced in MC-deficient mice as well as mice treated with the MC stabilizer cromoglycate [160]. Because proteomic profiling of ischemic brains indicated a suppression of pro-angiogenic proteins including Endoglin in MC-deficient mice compared to wild-type mice, the authors concluded that the absence of MC caused reduced angiogenesis [160]. Initial studies showed that TGF-β1 provokes chemotaxis of primary mouse and rat MCs [161]. In the respective study, the type I, type II and type III receptors were found to be expressed on the surface of murine MCs by cross linking experiments [161]. This was also recapitulated for chemotaxis in a human MC line HMC-1 and primary human MCs, in which all three isoforms of TGF-β provoked a chemotactic response and inhibit proliferation [151]. Nevertheless, in contrast to the rodent cells only type I and type II receptors, but not Endoglin, could be identified by cross linking studies in HMC-1 [151].

In a study analyzing primary human skin MCs it was shown that TGF-β has an impact on mediator release, apoptosis/proliferation and kit expression. The cells are of the connective tissue type MC_TC_ which characteristically express tryptase and chymase. It was reported that these cells express the type I, type II and both type III receptors, i.e., betaglycan and Endoglin, on mRNA level and surface expression of type I and type II receptor as assessed by flow cytometry [162]. In a more recent report, the use of Endoglin as a marker for human MCs was evaluated. Therefore, human tissue of a fibrous scar, neurofibroma, and tissues of patients with Mastocytosis were analyzed. The detection of MCs using standard methods including toluidine blue stain or immunodetection of tryptase was compared to immunodetection of Endoglin. Thereby, it was shown that Endoglin is highly expressed in MCs and localized to the secretory granules [163]. As outlined in more detail above, TGF-β1 has an impact on several key functions in the different types of MCs of different origins and the TGF-β co-receptor Endoglin is expressed in MCs. Currently, a functional link of Endoglin in TGF-β signal transduction in MCs is missing.

##### Lymphopoietic Cells

Endoglin and Leukemia

In order to analyze the role of Endoglin in leukemia cells and their tumor-initiating potential, human blasts from peripheral blood or bone marrow of acute myeloic leukemia (AML) and precursor B-acute lymphoblastic leukemia (ALL) patients were analyzed [164]. The majority of B-ALL and a high proportion of ALL blasts (47.5–98.5%) express Endoglin and the presence of Endoglin increased the leukemogenic potential *in vivo* as confirmed by Endoglin inhibition using TRC105 [164]. Nevertheless, the efficacy of TRC105 treatment was lower in case of the ALL cells. This was due to higher expression/plasma level of sEng in patients and the murine model causing titration of the antibody TRC105 [164,165]. In line, a recent flow cytometric analysis performed by Cosimato and colleagues showed that Endoglin is expressed in AML of different origin, B-cell precursor ALL and to a lesser extend in early T-cell precursor lymphoblastic leukemia [166].

In mouse peripheral blood about 2–5% of white blood cells express high amounts of Endoglin from which were 53% B-cells (B220^+^), 14% T-cells (CD3^+^) and 30% macrophages (Mac-1^+^). However, in HSC transplantation experiments it turned out that modulation of Endoglin expression does not alter the proportions of each of these lineages suggesting that white blood cell lineage choice in the adult is not influenced by Endoglin [100].

##### T-Cells

As mentioned above, MSCs have immunomodulatory function in regard to B-cells [81]. In addition, it has been delineated that also T-cell functions are regulated by MSC [167,168,169,170]. In a study addressing this topic on a mechanistic level, it was shown that adipose derived MSC (ASC) directly interacted with allo-activated peripheral blood mononuclear cells, including B-cells, NK-cells, and T-cell subsets. The bound CD4^+^ T-cells had a higher proliferative rate, activation state (CD25) and increased Endoglin expression compared to the unbound CD4^+^ T-cells. In agreement with higher Endoglin expression in those ASC-bound CD4^+^ T-cells, Smad1/5/8 was activated at a much higher level compared to unbound cells [171] (Figure 6A).

Endoglin expression was analyzed before in human peripheral blood cells and (low) expression could be demonstrated in monocytes (CD14^+^), some B-cells (CD19^+^/CD20^+^) and several subsets of T-cells including (among the CD4^+^ T-cells) memory and naïve T-cells, although other studies couldn’t detect Endoglin on peripheral lymphocytes [24,88]. Activation of CD4^+^ T (CD25^+^) cells by CD3 caused a time-dependent upregulation of CD105 at the cell surface, which was also seen in response to PMA treatment. Activation was accompanied by phosphorylation (Ser/Thr) of CD105 [24]. Modification of Endoglin engagement (overexpression or crosslinking) revealed that CD105 induces proliferation and counteracts the TGF-β1-mediated anti-proliferative effect and Smad activation as assessed in a reporter assay using a Smad3-sensitive (CAGA)_4_ reporter construct [172]. However, CD105 increased ERK1/2 phosphorylation in the absence or presence of CD3 activation [24] (Figure 6B). The expression of Endoglin in intratumoral Tregs (CD4^+^/CD25^+^/Foxp3^+^) was confirmed/shown in a recent study, which focused on the efficiency of TRC105/PD1 antibody treatment in different cancer settings [173]. In addition, co-localization of Foxp3 and Endoglin was also found in human colorectal cancer specimens. This co-expression in Treg is the basis for the efficiency of TRC105 in cancer treatment, leading to a reduction of intratumoral Treg and EC-dependent angiogenesis. These effects are mediated by CD8^+^ T-cells and the presence of FcγR leading to antibody-dependent cellular cytotoxicity (ADCC) [173]. In conclusion, CD105 is highest expressed in CD4^+^ T-cells, modifies their proliferation and offers the opportunity to eliminate intratumoral, immumosuppressive Tregs (Figure 6C).

##### B-Cells

As discussed, Endoglin was first identified by generating a panel of different antibodies against surface proteins of the pre-B-leukemic cell line HOON [119,174]. One of these antibodies (44G4) detected a polypeptide of ~180 kDa under nonreducing and ~95 kDa under reducing conditions that was characterized as Endoglin and which was highly expressed on the vascular endothelium [116,120,175]. However, this antibody was unreactive with B-cells, monocytes, granulocytes and resting or activated T-cells [120]. The antibody 29-G8 reacts with early B-lineage precursor cells (CD34^+^/CD19^+^) in fetal bone marrow and pro-erythroblasts (CD71^+^/Glycophorin A^+^) in adult bone marrow [174]. In addition, Endoglin was shown to be expressed in the cell lines HL-60 and KG-1a, which were derived from individuals with AML, in the cell line EM-3 derived from a patient with chronic myelocytic leukemia (CML) and in the B-lymphoblastoid cell lines 3104 and 3161 [120]. Therefore, Endoglin is expressed at variable low levels on cell lines of myeloid and B lymphoid origin. Analysis of patient’s primary cells also led to the finding that Endoglin (44G4) is predominantly expressed on leukemic cells of the B lymphoid and myeloid origin, but not by leukemic cells of the T lymphoid lineage [120]. 44G4 detected Endoglin expression in ALL cells derived from children with pre-B ALL and with AML but not in ALL cells of T-cell origin. Several cell lines of human pro-B and pre-B type were found positive for Endoglin expression [174]. Moreover, human leukemic cell lines, including pre-B- and precursor B-cell leukemic lines (NALM-6, SEMK2, and RS4;11) were positive for Endoglin. The more mature B-cells and T lymphoblastoid cell lines Raji and Jurkat lack Endoglin in line with data defining CD105 as a biomarker to discriminate between B- and T-lineage ALL [164,176]. The alternative Endoglin-specific antibodies 1G2 and 29-G8 did not cross-react with bone marrow B-cells and mature B-and T-cell, but with pro-B and pre-B leukemic cells [102,174]. In HOON, G2, and NALM-6 cell lines it was shown that Endoglin is recruited to a TGF-β1 receptor complex including TβRI and TβRII and therefore, it is obvious that Endoglin is engaged in active signaling in those cells [177].

## 4. Conclusions

In recent years, it has been anticipated that Endoglin expression/function is not restricted to ECs. Especially its differential expression during hematopoiesis argues for a role of this receptor in the development of individual cell lineages. This paves the way for a better understanding of hematopoiesis in general by using CD105 as a marker to purify and analyze individual developmental cell stages during hematopoiesis. In addition, it can be used as a marker to identify different types of leukemia relying on the occurrence of precursor cells in the peripheral blood. Moreover, Endoglin expression is present on mature immune cells of the innate, i.e., macrophages and MCs, and the adaptive, i.e., T-cells, immune system. Therefore, Endoglin might be a factor which helps to shape the immune response. Nevertheless, the data on Endoglin function in those mature, differentiated cells is still limited and needs further attention. Finally, the knowledge of Endoglin expression in individual immune cells might offer the opportunity to interfere with tumor promoting effects of immune cells and escape of cancers from immunosurveillance mechanisms as it has been shown for Tregs by using the anti-Endoglin antibody TRC105. In conclusion, the presence of Endoglin in immune cells should help to increase/provide the treatments and knowledge toward processes leading to cancer and immune escape as well as pathologies depending on vascular dysfunction associated with inflammatory processes.

## Figures and Tables

**Figure 1 ijms-21-09247-f001:**
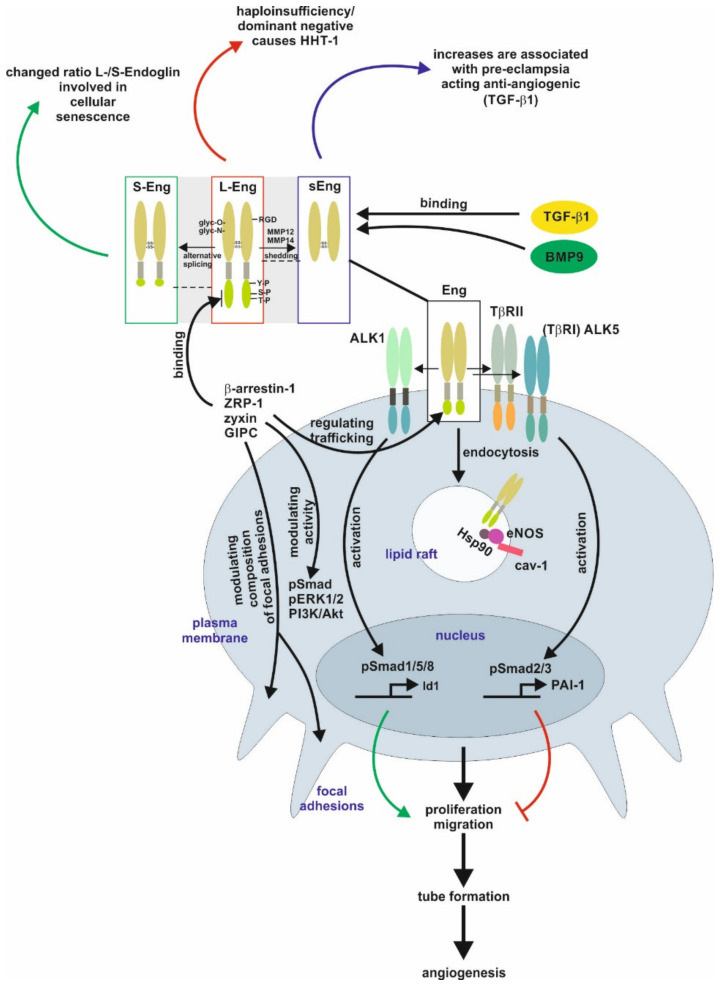
Endoglin biology in endothelial cells. (central part) Endoglin (Eng) is a component of a receptor complex in endothelial cells comprising the type I receptor dimers activin receptor-like kinase 5 (ALK5) or ALK1, and type II receptor dimers binding ligands, e.g., transforming growth factor-β1 (TGF-β1) and bone morphogenetic protein 9 (BMP9), of the TGF-β-superfamily. Interaction with these receptors either activates Smad1/5/8 or Smad2/3 signaling (phosphorylation) to regulate corresponding target genes, e.g., inhibitor of differentiation 1 (Id1) and plasminogen activator inhibitor-1 (PAI-1), and finally, cellular responses including proliferation and migration to balance the activation and resolution phase of angiogenesis. Localization of Eng in caveolin-1 (cav-1) positive lipid rafts mediates interaction of heat shock protein 90 (Hsp90) with endothelial NO-synthase (eNOS) regulating its function. (upper part) Beside the long variant of Eng (L-Eng), a C-terminally truncated, shorter splice variant (S-Eng) and a shedded form, mediated by matrix metalloproteinase (MMP)-14, comprising only the extracellular domain (sEng) has been identified. All three variants of Eng are involved individually in cellular (senescence) and pathological conditions (e.g., HHT-1, pre-eclampsia), respectively. The extracellular domain of Eng is *N*- and *O*-glycosylated and human Eng contains an integrin binding motif (RGD) motif. The L-Eng C-terminal domain is a substrate for TGF-β receptors leading to serine/threonine phosphorylation. Tyrosine phosphorylation is mediated by Src. L-Eng interacts with its cytoplasmic domain with several intracellular proteins regulating receptor trafficking, phosphorylation/activity of kinases, and focal adhesion amongst others. The functional difference of S-Eng compared to L-Eng most likely arises from the missing binding- and phosphorylation-sites in its C-terminus. Abbreviations used are: GIPC, GAIP C-terminus-interacting protein; HHT-1, hereditary hemorrhagic telangiectasia 1; Smad, small mothers against decapentaplegic; pERK1/2, phosphorylated extracellular signal-regulated kinase 1/2; PI3K/Akt, phosphatidylinositol 3-kinase; TβRI/II, transforming growth factor-β1 type I or II receptor; ZRP-1, zyxin-related protein-1.

**Figure 2 ijms-21-09247-f002:**
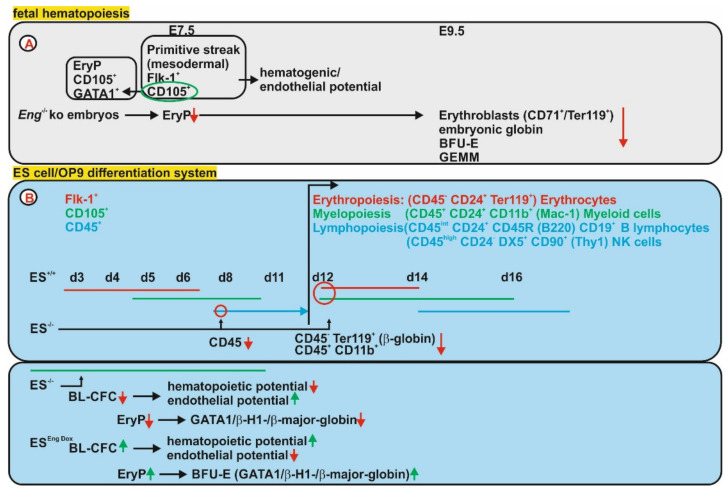
Endoglin expression and function in fetal hematopoietic cells. (**A**) Endoglin (Eng) expression is detected as early as embryonic day 7.5 (E7.5) in mesodermal cells (Flk-1^+^) and primitive erythroblasts (EryP, GATA1^+^) of the blood islands in wild-type mice. Endoglin expressing cells comprise all hematogenic and endothelial potential. In line, Eng deficiency (Eng^−/−^) resulted in reduced EryP colonies and at E9.5 reduced erythroblasts (CD71^+^/Ter119^+^) in the yolk sac (YS) and diminished numbers of BFU-E and GEMM progenitors. (**B**) (upper) Initially it was shown in the development of embryoid bodies, by culturing ES cells on OP9 feeder cells, that Endoglin is transiently expressed during the progression from Flk-1^+^/CD45^+^ to the Flk-1^−^/CD45^+^ stage. Using Endoglin deficient ES cells it gets evident that although Flk-1 mesodermal precursor differentiation is normal, the CD45^+^ subset of hematopoietic cells is reduced at d9. In addition, the differentiation of ES cells into erythroid (CD45^−^/Ter119^+^) and myeloid (CD45^+^/CD11b^+^) was severely diminished along with a strongly reduced adult β-globin gene expression indicating impaired definitive erythropoiesis. (lower) Another study showed CD105 expression already in ES cells and Eng deficiency affected the blast-colony forming cell (BL-CFC) frequency and lowered its hematogenic potential, while increasing endothelial potential. EryP frequency was reduced in line with reduced expression of GATA1, embryonic and adult globin. The afore mentioned results were confirmed by using ES cells with doxycycline-inducible overexpression of Eng which leads to revers effects in comparison to the above described hematopoietic defects.

**Figure 3 ijms-21-09247-f003:**
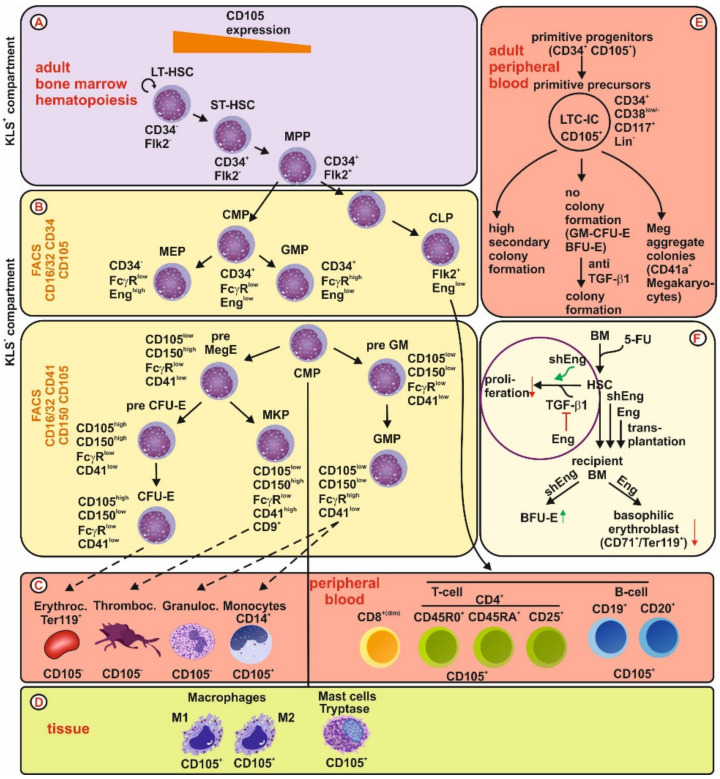
Endoglin expression and function in adult hematopoietic cells. (**A**) Bone marrow: In the adult KLS^+^ (Kit^+^/Lin^−^/Sca-1^+^) compartment, Eng is highest expressed in long term repopulating hematopoietic stem cells (LT-HSC). Expression in short term (ST-HSC) and multipotent progenitors (MPP) is much lower. (**B**) In the adult KLS^−^ (Kit^+^/Lin^−^/Sca-1^−^) compartment, Eng expression is low in the common myeloid (CMP) and lymphoid progenitors (CLP). Of the committed precursor cells, CD105 is low expressed in the granulocyte-monocyte progenitor (GMP) but highly expressed in the megakaryocyte-erythroid progenitor (MEP). Using a different FACS panel including CD105, it could be shown that Endoglin is transiently high expressed in pre-CFU-E and CFU-E. (**C**) peripheral blood: In adult circulating cells, CD105 could not be phenotypically detected in erythrocytes, thrombocytes and granulocytes. However, weak expression was shown in CD14^+^ monocytes and subsets of T- and B-cells. (**D**) Of the tissue residing cells, CD105 is expressed in M1 and M2 polarized macrophages as well as in tryptase positive mast cells. (**E**) A subset of circulating CD34^+^ co-expresses CD105. Those progenitor cells contain primitive precursors with a high frequency of long-term culture initiating cells (LTC-IC). In addition, these cells show no colony formation, except megakaryocyte colonies, unless TGF-β1 is blocked. (**F**) By treatment of bone marrow cells with 5-fluorouracil (block of cell cycle), HSC are enriched. Manipulation of Eng expression in those myeloid cells, i.e., shRNA knock-down or lentiviral overexpression, showed that Eng blocks TGF-β1-mediated inhibition of proliferation. Moreover, analysis of recipient bone marrow indicated that Eng has no impact on white blood lineages but has a negative impact on erythroid burst-forming unit (BFU-E) formation and erythroid differentiation (reduction of basophilic erythroblasts, CD71^+^/Ter119^+^).

**Figure 4 ijms-21-09247-f004:**
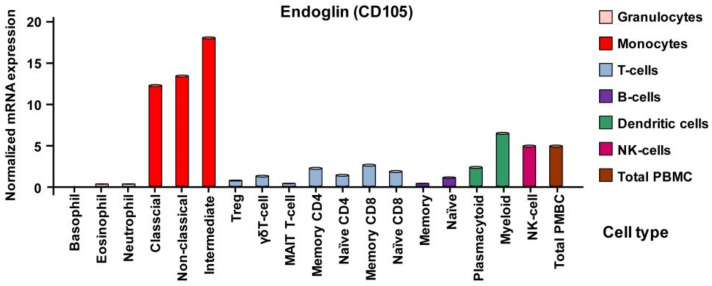
Transcript analysis of endoglin expression in peripheral blood cells. Transcript analysis of peripheral blood cells verified the very low presence of *endoglin* in granulocytes, low expression in T- and B-cells and showed a high expression in monocytes. In addition, dendritic cells express considerable amounts of *Eng* mRNA. Expression data were taken from the entry for Endoglin in the Human Protein Atlas [101].

**Figure 5 ijms-21-09247-f005:**
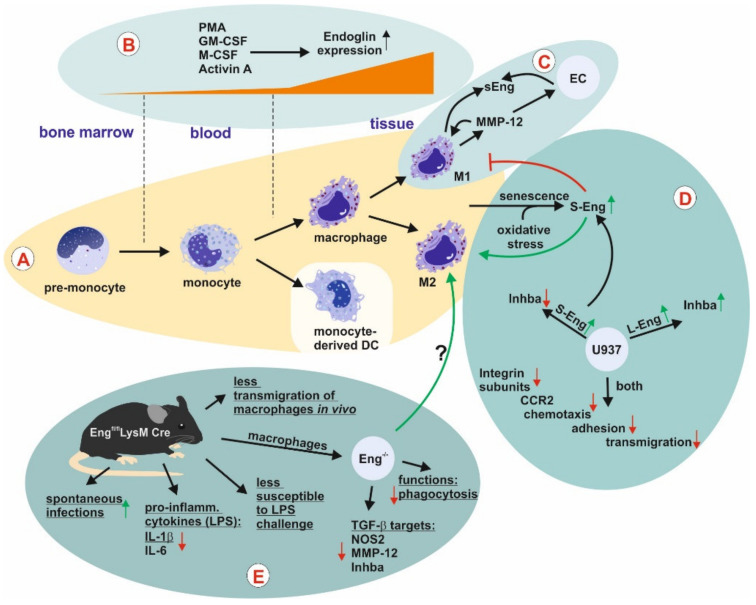
Endoglin expression and function in macrophages. (**A**) Bone marrow-derived pro-monocytes differentiate to monocytes which enter the circulation. (**A**,**B**) Monocytes extravasate, enter tissues, differentiate to macrophages, and are polarized to anti- (M2) or pro-(M1) inflammatory macrophages upon the appropriate stimuli, of which both express Endoglin (Eng). In the course of these processes L-Eng is upregulated. Monocyte-derived dendritic cells have not been analyzed for Eng expression so far. Cellular aging and oxidative stress causes senescence accompanied by proportional higher upregulation of S-Eng. (**C**) M1 polarized (pro-inflammatory) macrophages are pro-inflammatory and anti-angiogenic. This is in part due to the secretion of MMP-12 which is missing in the secretome of M2 macrophages. Secretion of MMP-12 causes shedding of Eng in the membranes of macrophages and EC leading to the liberation of sEng, which is an anti-angiogenic molecule. (**D**) For functional analysis, U937 L- and S-Eng transfectants were analyzed by proteome [135] and transcriptome [133] analysis. In line, both studies showed a reduced expression of different integrin subunits in both transfectants accompanied by reduced adhesion. However, the data of Aristorena and colleagues [135] suggested that S-Eng blocks M1 and favors M2 polarization, whereas the L-Eng data were not that clear. Both studies showed in accordance that L-Eng up- and S-Eng down-regulates the inhibinBα (activin A) subunit. (**E**) A myeloid lineage-specific Eng knockout showed spontaneous infections and a lower inflammatory response *in vivo* coupled to compromised inflammatory macrophage functions *in vitro*, implying less M1 activity in the absence of Eng. Abbreviations used: CCR2, C-C chemokine receptor type 2; DC, dendritic cell; EC, endothelial cell; Eng, Endoglin; GM-CSF, granulocyte-macrophage colony-stimulating factor; IL, interleukin; LPS, lipopolysaccharide; M-CSF, macrophage colony-stimulating factor; MMP, matrix metalloproteinase; NOS2, inducible NO-synthase; PMA, Phorbol-12-myristate-13-acetate.

**Figure 6 ijms-21-09247-f006:**
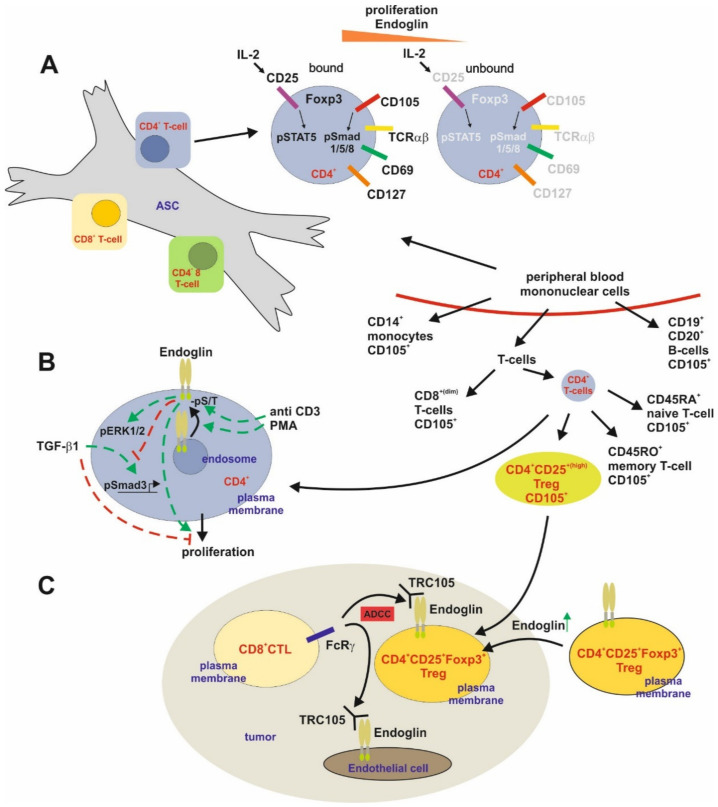
Endoglin expression and function in T cells. (**A**) Adipose tissue derived stem cells (ASC) are able to regulate T-cell dependent immune responses in part by sequestering reactive CD4^+^ T-cells. Allo-activated CD4^+^ T cells tightly bound to ASC show a higher CD25 expression and in turn IL-2-dependent STAT5 activation. These cells present a higher proliferation and increased Endoglin (Eng) expression along with a higher Smad1/5/8 phosphorylation (activation). Nevertheless, aside the higher Foxp3 expression, the bound cells also had a higher CD127 expression excluding that they are Tregs. (**B**) High expression of Eng is present in CD4^+^ T-cells isolated from peripheral blood. TCR/pathway activation of these CD4^+^ T-cells by CD3 or PMA leads to a higher surface expression of Eng and increased serine/threonine phosphorylation of the receptor. Activation of Eng induces ERK1/2 phosphorylation and higher proliferation. On the other hand, Eng interferes with TGF-β1-mediated Smad signaling and inhibition of proliferation. (**C**) In the setting of tumorigenesis, intratumoral CD4^+^ T-cells, i.e., CD25^+^/Foxp3^+^ Tregs, express higher Eng compared to their circulating counterparts. This causes an antibody-dependent cellular cytotoxicity (ADCC) resulting in reduction of intratumoral Tregs (immunosuppressive) upon application of TRC105 *via* CD8^+^ T-cells.

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
