# Peer review of "Endoglin: An ‘Accessory’ Receptor Regulating Blood Cell Development and Inflammation"

_ijms, 2020, doi:10.3390/ijms21239247_

Round 1

Reviewer 1 Report

In this review article, Meurer and Weiskirchen describe the role and functions of endoglin in hematopoiesis and inflammation. The article offers an exhaustive panoramic view of CD105 involvement in stem cell biology, hematopoiesis and immunology.

The article is well written and carefully compiled and I believe it is a good resource for the readers in the aforementioned fields.

There are few aspects that have not been touched and I believe could be useful for a better definition of CD105 regulation. I would suggest the authors to describe the epigenetic mechanisms of CD105 expression (epigenetic silencing, etc.) as well as the post-translational regulation of the protein.

Since the authors extensively describe the roles of CD105 across hematopoietic differentiation, I suggest adding a simple chart that tracks the expression of CD105 across early and mature populations. Figure 4 in part does this, but I would extend it to immature populations and I would complement it with expression data for TGF-b1 and maybe some selected differentiation markers (CD34, ALK1, ALK5, CD71, etc).

Author Response

Dear Reviewer 1,

please see our point-by-point response to your comments in the attached pdf-File.

Regards

Reviewer 2 Report

The review “Endoglin, an ’Accessory’ Receptor Regulating Blood  Cell Development and Inflammation” by Meurer  and Weiskirchen is a very complete and extensive work review on the role of Endoglin in the context of the immune response and blood cell biology.

This review is too long, fact that is a virtue and a flaw: it covers most papers published on this subject, and explains all of them with detail; therefore, the reading is slow.

The figures are as well too complex but at the same time very complete.

It would be great if authors could simply somehow the manuscript, both the text and the figures. Otherwise, authors should guide the readers with summaries and conclusions somehow highlighted throughout the text.

The references cannot be reviewed as there is a mistake in the numbering.

Some parts of the text should be re-ordered to facilitate the comprehension of the reader.

  1. ENG in general,
  2. ENG in disease (HHT, pre-eclampsia, cancer)
  3. ENG in the hematopoietic system
  4. Conclusion

The first part of the manuscript, (36-104) is not very clear. It is not well explained what are the function of ENG, what TGF-b superfamily ligands does modulate, how this modulation occurs in endothelial cells.

Paragraph 106-118 is not clear if it refers to endothelial cells or to hematopoietic cells or it is a general comment.

The title of part 4 “Historical Point of View” does not really match with content.

Author Response

Dear Reviewer 2,

please see our point-by-point response to your comments in the attached pdf-File.

Regards
